# Longitudinal Analysis of Teacher Technology Acceptance and Its Relationship to Resource Viewing and Academic Performance of College Students during the COVID-19 Pandemic

Rubia Cobo-Rendon [1], Karla Lobos Peña [1,*], Javier Mella-Norambuena [1,2], Nataly Cisternas San Martin [1] and Fernando Peña [1]

1   Laboratorio de Investigación e Innovación Educativa IDEClab, Dirección de Docencia, Universidad de Concepción, Concepción 4030000, Chile; rubiacobo@udec.cl (R.C.-R.); javimella@udec.cl (J.M.-N.); ncisternas@udec.cl (N.C.S.M.); fernpena@udec.cl (F.P.)

2   Programa de Doctorado Educación en Consorcio, Universidad de Católica de la Santísima Concepción, Concepción 4030000, Chile

\*   Correspondence: karlalobos@udec.cl

**Abstract:** Due to COVID-19, teachers quickly changed their courses from traditional face-to-face modality to emergency remote teaching (ERT), relying on learning management systems (LMS). In this simple prospective design study, we analyzed the relation of the level of teachers' technological acceptance at the beginning of ERT (March 2020) considering three variables: the time spent by teachers in the LMS during that semester, the percentage of LMS's resources their students viewed during the semester, and the final academic performance of the same students at the end of that semester (September 2020). This study included 251 teachers (57% male) and 12,185 students (45% male). We measured the teachers' level of acceptance with the Spanish version of the Questionnaire Technology Acceptance Model (TAM). We found that the relation between the teacher's acceptance and their time spent on the LMS was significant and positive (rho = 0.24, $p < 0.001$). In addition, teachers' perception of LMS's easiness is related to the percentage of educational resources their students utilized (rho = 0.26, $p < 0.001$). Finally, we found a relation between the usefulness dimension of the TAM to the academic performance of the students at the end of that semester (rho = 0.18, $p < 0.01$). Considering these results, we discuss practices for implementing quality education.

**Keywords:** TAM Model; learning analytics; academic performance; higher education; COVID-19

## 1. Introduction

Due to the rapid changes produced worldwide by the COVID-19 pandemic in educational institutions [1], teachers at all educational levels had to make changes in the way they taught, relying on the use of technology [2,3]. Specifically, university teachers had to quickly transition from face-to-face teaching to emergency remote teaching (ERT) in order to assure academic continuity [4,5]. ERT is characterized by being developed in virtual environments with urgent and scarce planning with the goal of maintaining the continuity of educational training during a crisis [5].

In this scenario, teachers were faced with the untimely need to quickly modify their programs to an online modality. University institutions also had to make decisions and favor this modality by incorporating virtual tools and learning management systems (LMS) [6,7]. However, for these tools to play a positive role in the quality of teaching, teachers require new skills and readiness for the use of technological resources that allow them to take full advantage of their benefits [8].

*1.1. Technology Acceptance Model (TAM)*

In recent decades, several theoretical models have been proposed that seek to evaluate the willingness to use technological resources and the processes of acceptance and use of technology by teachers. Among them is the technology acceptance model TAM proposed by Davis (1989). Its objective is to explain the reasons that users have for using technologies [9]. This model has become one of the most interesting for researchers and technology designers in recent years, being used in various contexts where technology is implemented; for example, in the evaluation of the mobile experience [10,11], in the implementation of technology in health areas [12] and in the use of technological devices [13]. The TAM model has been undisputedly the most tested and validated model in different contexts and studies, confirming its robustness and ability to predict technology adoption in users [14,15].

This model is explained from the Theory of Reasoned Action [16]. This theory is based on the premise that the perception of a person will determine his/her attitude and behavior, i.e., the attitude and behavior that a person may have about the use of a particular technology is influenced by the perceptions that the person has about it [17]. In this sense, in the approach proposed by Davis, the level of use of a technology depends on the users' perceptions regarding the ease and usefulness of this technology for the performance of a particular task [18]. These two aspects (ease and usefulness) influence the user's attitude and intention to accept a technological system or device [15]. Therefore, these factors are two of the main extrinsic motivations (beliefs) that influence an individual's acceptance and use of technology. The two variables are positively correlated (i.e., a user-friendly website is more likely to be perceived as useful) and both influence an individual's attitude toward the use of a given technology. Likewise, under this model, attitudes toward technology use are positively correlated with the behavioral intention to use it, which, in turn, is positively correlated with actual use [18]. All this makes the TAM a powerful tool for describing technology adoption by teachers in higher education [15].

The use of technology for teaching and learning processes is usually referred to as digital learning [19]. In this context, technology or educational technology involves implementing technology-based software or applications for delivering learning resources and implementing learning activities in face-to-face, online or hybrid classrooms [20]. The role of educational technology in the COVID-19 pandemic has been fundamental. Teachers could implement the emergency remote teaching modality thanks to learning management systems LMS, like Moodle or Canvas, and videoconferences systems like Zoom or Teams. These two technologies are among the most used during the pandemic period. However, once educational institutions start implementing face-to-face modality again, it is expected to observe new technologies that satisfy the blended and hybrid modalities [21].

In the educational setting, a systematic review analyzed published research from 2003 to 2018 on the application of TAM including 71 papers, [22]. Indicated that TAM is a credible model to facilitate the evaluation of various learning technologies [22]. The core variables of TAM, perceived ease of use and perceived usefulness, affect the acceptance of learning with technology [22]. Researchers also reported that a small percentage of the articles analyzed (6%) focused on teachers [22]. In another systematic review regarding teachers and students [23], researchers analyzed 50 publications that aimed to investigate the acceptance of technology at that level. In the results, the authors indicate that most of the research using the TAM model refers to education linked to information and communication technologies and how this model could identify relevant elements from the point of view of teachers as well as students [23].

A meta-analysis developed on the TAM model in teachers, particularly for pre-service and in-service teachers, sought to synthesize 124 correlation matrices from 114 empirical TAM studies ($n$ = 34,357 teachers) to test the fit of the TAM and its versions. They analyzed data from teachers in schools, universities, or technology acceptance in educational contexts, most of the participants were from Asia or America. The authors found that the TAM explains well the acceptance of technology in teachers [15].

In the case of university teachers, some studies on TAM report that the measurement and evaluation of this model is beneficial for the prediction of teachers' intentions of implementing learning management systems [24,25]. In the same context, recent reports regarding the use of LMS in higher education institutions, in which responses from both students (*n* = 584) and teachers (*n* = 42) were analyzed, reported different perceptions between students and teachers regarding the use of LMS, so particular research is necessary considering the type of user [25].

Now, faced with the context of the ERT due to the pandemic, in order to study how the TAM model research was used in this scenario, we conducted a brief systematic review following the PRISMA model [26]. For this case we reviewed the Web of Science (all collections), EBSCO (all collections) and Scopus databases, using the keywords of "technology acceptance" AND "COVID-19" OR "coronavirus" OR "2019-nCoV" OR "SARS-CoV-2" OR "COV-19" AND "higher education" OR "college" OR "university". A total of 30 studies were identified. In phase 2, 17 duplicate papers were eliminated, identifying a total of 13 original investigations. In phase 3, We are selected 11 papers what included the keywords in their title and abstract, of which eight full manuscripts were accessed. Finally, in phase 4 we used the selection criteria (research in the context of COVID-19, on technological acceptance in higher education), concluding with a sample of eight papers (See Supplementary Material).

From the analysis, we found that three studies seek to assess specific elements of online education of the TAM model, such as the use of Zoom, Microsoft Teams and Google classroom as videoconferencing platforms [27–29]. The remaining papers focused on online education in general [30–34]. Regarding the research participants, most were university students (*n* = 7) and only one research focused on higher education instructors or teachers [34]. The latter is qualitative, and its authors describe some exploratory findings on the level of acceptance of the teachers. Teachers stated that virtual tools were useful to promote the fulfillment of the objectives of their courses, the realization of evaluative processes, and instructional methods. Nevertheless, they did not neglect the challenges that ERT represented in terms of interaction, evaluation, and development of their pedagogical practices [34].

In conclusion, the TAM model (a) is useful for the evaluation of the level of technological acceptance with respect to specific technological tools and also with respect to the online teaching modality in general in the context of ERT, (b) is positively associated with the satisfaction and participation of users in digital platforms, (c) it successfully explains the factors that predict the use of e-learning and, (d) allows to distinguish different user roles in regarding technology (such as student and teacher).

### 1.2. Educational and Learning Analytics in Higher Education

In higher education, the collection and use of university data has expanded dramatically over the past two decades [35]. One of the ways to assess how faculty technology acceptance beliefs impact the educational experience of students in virtual learning environments is through the exploration of data that describes the behavior of users in online platforms. This is known as academic analytics and learning analytics.

On one hand, academic analytics refers to digital information that is used to manage an institution to benefit data-driven decision-making [36]. These analytics are nourished by the information provided by institutional systems about undergraduate programs, subjects, teachers, and students [37]. On the other hand, learning analytics refers to the process of measuring, collecting, analyzing, and reporting student-centered data with the goal of improving the student's educational experience [36].

Understanding teacher factors that might drive student academic performance is becoming an important topic of research in Educational Psychology. In general terms, both researchers and educational authorities are interested in identifying elements that could contribute to improve the quality of education [38]. In this sense, the United Nations included education as one of the relevant topics within the sustainable development

objectives. Among the specific objectives proposed, the aim is to promote safe and effective learning environments for all students, as well as improving teaching skills [39]. The use of data provided by academic and learning analytics will allow the implementation of effective strategies to improve the quality of education. In this sense, analytics are a powerful mechanism to help students, instructors, teachers, designers, and developers of learning systems to better understand educational processes and predict students' needs and performance [36,40].

Academic and learning analytics enables the improvement of teaching and learning processes through the analysis of educational data available in the LMS [40], which are software developed exclusively to manage the teaching and learning process [25]. From an LMS, a digital fingerprint can be obtained that describes the behavior both teachers and students (connection time, access to resources, participation in forums, etc.) [41]. One of the most widely used LMS in Higher Education is Canvas In structure LMS. It was created in 2010 and it is currently considered the fastest growing LMS worldwide [42]. It is used by about 30% of higher education institutions in the United States [43]. Research on LMS usage by faculty indicates that compared to other LMSs, Canvas was the most popular system. Moreover, research reports that teachers perceived ease of use and user satisfaction. The features of this LMS reported as relevant are related to the possibilities of structuring the class, creating assignments, uploading files, and other aspects that distinguish it other LMSs [44,45].

### 1.3. The Present Study

The COVID-19 pandemic forced universities to change the ways in which they taught. In this case, online education was considered as an effective sustainable learning solution in the conditions of ERT and in the conditions that may arise in the future [46,47]. The sustainable development is defined as meeting the needs of current generations without compromising the ability of future generations to meet their needs [48]. The online education focuses on promoting student participation, encouraging critical analysis, regulation of their learning pace and other elements of the educational experience, thus satisfying the needs of each student [2,49,50]. However, to be able to respond to this, one of the factors that guarantees success in the development of online education of quality is the paper of the teacher [47,51].

Understanding the processes of acceptance of technology by university professors is a relevant task because it allows a better understanding of the possible mechanisms that exist for the implementation of e-learning in higher education [15]. From sustainable development's perspective, university institutions need to implement strategies to transform the way in which education is it developed today [52]. In this sense, for teachers to be able to implement digital resources such as LMS during university training processes, it is necessary that they possess beliefs of usefulness and ease of use of these tools, influencing their pedagogical practices [53].

Additionally, analyzing the process of acceptance of technology also allows evaluating how teachers' beliefs could impact the development of quality learning experiences in students. The present study aims to contribute to the proposals of the various declarations on sustainable development regarding the importance of universities being able to provide teachers with resources that contribute to the education of their students and, therefore, to society [48,52].

For this reason, the general objective of this research is to analyze the relationship between the level of technological acceptance of teachers at the beginning of the ERT semester (T1) with (1) the time spent by teachers in the Canvas LMS, (2) the percentage of resources viewed their students in the LMS, and (3) the final academic achievement of the same students at the end of that semester (T2).

To this end, we set the following specific objectives:

1.  To describe and relate the level of technological acceptance of teachers measured at the beginning of the academic period (T1) and the time spent by teachers in the Canvas LMS at the end of the ERT due to COVID-19 (T2).
2.  To analyze the relationship between the level of technological acceptance of teachers (T1) with the percentage of resources viewed, and the academic achievement obtained their students at the end of the ERT semester due to the COVID-19 pandemic (T2).

    We propose the following hypotheses:

**Hypothesis 1 (H1).** *Teachers with higher levels of technology acceptance at (T1) will have greater time spent on the Canvas LMS at the end the ERT semester (T2).*

**Hypothesis 2 (H2).** *The teacher's level of technological acceptance (T1) is positively related to time spent on the Canvas LMS at the end the ERT semester (T2).*

**Hypothesis 3 (H3).** *The teacher's level of technological acceptance (T1) is positively related to the percentage of resources viewed their student in the LMS Canvas at the end the ERT semester (T2).*

**Hypothesis 4 (H4).** *The teacher's level of technological acceptance (T1) is positively related to the student's academic performance at the end of the ERT semester (T2).*

Teachers' pedagogical beliefs play a key role in their pedagogical decisions about how to integrate technology into their classroom practices and how to do so [54]. Some research shows that when teachers present high levels of technological acceptance with an LMS, the possibility of using it increases [18,24,25], evidence that supports H1. In addition, it has also been shown that teachers' beliefs can intervene in the educational experiences and performance of students during the development of their courses [55], evidence that we seek to deepen with H2 and H3.

## 2. Materials and Methods

The present research corresponds to a simple prospective non-experimental design [56]. We analyzed the TAM model in teachers who initiated the design and development of ERT courses due to the COVID-19 pandemic (T1). We related this variable to the teacher's time in the Canvas LMS (T2), to the students' level of visualization of educational resources, and the academic performance obtained by students at the end of the courses (T2).

### 2.1. Participants

A total of 251 university teachers participated in this study. These belonged to a public university in Chile. From the total of participants, 143 were male (57%), with a mean age of 48.17 years (SD = 11.17). Of the teachers participating in this study, 126 (50.2%) have a doctorate degree and 79 (31.7%) have a master's degree or a specialty degree. The scientific area where they teach, the years of teaching experience and the type of teaching day we are described in Table 1. These teachers designed and implemented a total of 487 online courses during the ERT due to COVID-19.

From learning analytics, it was possible to obtain the percentage of interaction and academic performance of 12,185 university students who had taken classes with any of the teachers in the sample and that agreed to participate in the study. From the total of students, 6703 were women (55%) and the average age was M = 22.96 (SD = 2.82). From the total of students, 23.62% were students in their first academic year. Table 2 shows the description of the sociodemographic variables of the students according to the scientific area to which they belong.

**Table 1.** Descriptive statistics on the area of performance, years of experience and time of dedication of the participating teachers.

| Area OECD | *n* | Age M (SD) | Years of Experience M (SD) | Type of Working Day Full-Time | Part-Time |
|---|---|---|---|---|---|
| Natural Sciences | 81 | 48.93 (11.9) | 6.84 (7.85) | 66 | 15 |
| Engineering and Technology | 27 | 43.56 (10.66) | 8.73 (7.84) | 19 | 8 |
| Medical and Health Sciences | 40 | 44.35 (9.31) | 6.05 (5.73) | 25 | 15 |
| Agricultural Sciences | 25 | 56.08 (9.56) | 11 (7.56) | 20 | 5 |
| Social Sciences | 57 | 49.14 (11.44) | 9.1 (8.61) | 26 | 31 |
| Humanities | 21 | 46.43 (7.41) | 6.05 (6.23) | 13 | 8 |

Note: M: Median; SD: Standard Deviation; *n* = 251.

**Table 2.** Descriptive statistics of the sociodemographic variables of the student's participants according to the scientific area to which they belong.

| Area OECD | *n* | Sex Female | Male | Age M (SD) | Academic Year 1st | 2nd | 3rd | 4th |
|---|---|---|---|---|---|---|---|---|
| Natural Sciences | 2795 | 2044 | 751 | 21.52 (2.26) | 668 | 591 | 509 | 1027 |
| Engineering and Technology | 2434 | 656 | 1778 | 21.39 (2.45) | 604 | 596 | 488 | 746 |
| Medical and Health Sciences | 3700 | 2243 | 1457 | 22.04 (3.42) | 884 | 769 | 843 | 1204 |
| Agricultural Sciences | 1411 | 664 | 747 | 22.26 (3.2) | 295 | 330 | 266 | 520 |
| Social Sciences | 1412 | 788 | 624 | 22.62 (3.21) | 303 | 160 | 220 | 729 |
| Humanities | 392 | 280 | 112 | 21.77 (2.84) | 104 | 78 | 87 | 123 |

Note: M: Median; SD: Standard Deviation; *n* = 12,185.

*2.2. Instruments*

2.2.1. TAM Model in Teachers: Perceived Usefulness and Ease of Use

A Spanish version of the Measurement Scales for Perceived Usefulness and Perceived Ease of Use (TAM) designed by Davis (1989) was used to evaluate people's perception of usefulness and ease of use of devices or software in digital environments. This scale consists of 12 items distributed in two dimensions: perceived usefulness and perceived ease. The former has six items oriented to evaluate the user's perception of the benefit of the device or software in improving productivity and performance of their work. The latter is composed of six items which aim to evaluate the perception of competence in the use of the device or software and the ease of integration with the user's work. Both dimensions are answered on a Likert-type scale with five response options ranging from strongly disagree = 1 to strongly agree = 5. Scores above 3 are considered as a high level of technological acceptance. In the English version, adequate internal consistency indices were identified ($\alpha$ = 0.98 for usefulness and $\alpha$ = 0.94 for ease of use) as well as high convergence, discriminant, and factorial validity (Davis, 1989). For this study, adequate internal consistency indices were found ($\alpha$ = 0.93 for usefulness and $\alpha$ = 0.93 for ease of use and $\alpha$ = 0.94 for the complete scale). We confirmed an adequate fit of the model with two factors proposed by the authors of the scale ($X^2(53)$ = 154.647, $p < 0.001$; CFI = 0.956; TLI = 0.945; SRMR = 0.050; RMSEA = 0.078 [0.064, 0.092], $p < 0.001$.

2.2.2. Teacher Academic Analytics: Time Spent

For the evaluation of teacher behavior in the Canvas LMS, we analyzed the variable of time spent on platform from the academic analytics. For this purpose, the teacher's log records were extracted [57]. With this data we worked with two variables:

Average session time: where each session is calculated as the time between interactions (or events associated with a timestamp) in the Canvas LMS, with a cut-off point of ten minutes. If the user, in this case the teacher, does not perform any action within ten minutes, the session is finished. This is used to calculate the average time spent on platform by teachers per session.

Average session time: to have a standardized measure of the connection time considering the characteristics of the courses, we added the duration of all sessions and divided it by the number of credits SCT of each subject taught by the teacher during the ERT 2020 period. Thus, the calculated time is assigned to 1 credit. The SCT credits are designed based on the estimated time dedicated by the student to achieve the learning outcomes. One SCT credit is equivalent to a range of workloads between 24 and 31 h by the student [58].

### 2.2.3. Student Learning Analytics

In the case of student analytics, we extracted the data associated with the number of visualizations of educational resources that were available in virtual classrooms of the participating teachers' courses. For this purpose, the information from the students' logs was analyzed. This information was used to generate the variable of percentage of resources viewed:

The percentage of resources viewed: for each course, we considered the resources with which at least one student interacted, this gave us the total number of resources. Then, for each student, we obtain the number of resources that they interacted with. Finally, the percentage is calculated as the number of resources that students interacted with over the total number of resources on platform.

### 2.2.4. Academic Achievement

The grade point average (GPA) in the first semester of 2020 was obtained from the academic record of the university and considered as the students' academic performance. In Chile, the GPA is constructed on a scale from 1.0 to 7.0 points. The grades from 6.0 to 7.0 correspond to an academic performance considered as "excellent." The grades from 5.0 to 5.9 are labeled as "good" grades, while 4.0 to 4.9 are defined as "satisfactory." Last, grades from 1.0 to 3.9 are "unsatisfactory," which means the student failed the course [59].

### 2.3. Procedure

This research was endorsed by the Ethics Committee of the participating university, corroborating the ethical criteria for research with human beings. The application of the instrument on the TAM model of the teachers was carried out in digital format by sending it to the institutional e-mails in a single opportunity. The link to answer the questionnaires was available during the first three weeks of March 2020 (beginning of the first ERT academic period). Participating teachers responded to the questionnaire after reading and signing the consent form. Information on teacher academic analytics and student learning analytics were obtained at the end of the semester from the Canvas platform. The students' academic performance was obtained from the university's official information recording platforms. At the beginning of the semester, the participating students signed a consent form authorizing access to their academic information for the development of the research.

### 2.4. Statistical Analysis

To obtain the information on the academic analytics of teachers and student learning in the Canvas LMS, the Canvas Data portal service was used. The information was obtained from the Log table or request table. Python programming language was used, using an algorithm to review the information provided by the URL. We used the information provided by the "TimeStap" to identify the time of interaction of the teacher with the Canvas LMS. For the analysis of the resources viewed by the students we used regular expressions to identify the educational resource linked to the URL.

Descriptive analyses were performed on the study variables. The assumption of normality was verified. In this case we used the Kolmogorov-Smirnov test with the Lilliefors modification since the total sample size of teachers was larger than 50 participants. We applied the Levene's test to verify the constant variance between groups (homoscedasticity) [60].

Since the assumption of normality was not met, nonparametric statistical procedures were used. Spearman's rho was used to evaluate the relationships between variables. For the evaluation of differences by groups of high and low technological acceptance, we used Yuen's robust test and the one-way ANOVA test for trimmed means for statistical analyses employing more than two groups [61,62]. The method proposed by Algina et al. (2005) was used for the effect size analysis of the results [63].

In the case of the analysis for the low technological acceptance group, since it had fewer than 50 participants, normality was verified by means of the Shapiro Wilk test [64]. It was possible to verify that the data in all cases did not follow a normal distribution ($p < 0.001$). The data analysis was made with R software version 4.1.0 (18 May 2021).

## 3. Results

The general objective of this work was to analyze the relationship between the level of technological acceptance of teachers at the beginning of the ERT semester (T1) with (1) the time spent by teachers in the Canvas LMS, (2) the percentage of resources viewed their students in the LMS, and (3) the final academic achievement of the same students at the end of that semester (T2). To address this objective, we will present the results according to the specific objectives described in the Section 1.3.

### 3.1. Technological Acceptance of the Canvas LMS and its Relationship to Time Spent Teacher during ERT by COVID-19

To respond to the first specific objective, we describe in Table 3 the measures of central tendency and dispersion of the variables studied. We identify a high level of technological acceptance by teachers regarding the use of the Canvas LMS (average > 3). At a specific level, the results indicate that, on average, teachers perceived greater ease (M = 3.82) than perceived usefulness (M = 3.63) concerning the Canvas LMS.

**Table 3.** Descriptive statistics on technological acceptance and academic analytics of teachers during ERT due to COVID-19.

| Variables | Min | Max | M | SD | Mdn | Asymmetry | Kurtosis |
|---|---|---|---|---|---|---|---|
| Perceived usefulness | 1 | 5 | 3.63 | 0.86 | 3.7 | −0.37 | −0.07 |
| Perceived Ease | 1 | 5 | 3.82 | 0.86 | 4.0 | −0.72 | 0.26 |
| Technological Acceptance | 1 | 5 | 3.72 | 0.77 | 3.8 | −0.55 | 0.25 |
| Platform connection time * | 0.01 | 145.03 | 13.36 | 17.82 | 8.0 | 3.65 | 18.92 |
| Average session time ** | 0.01 | 1.7 | 0.27 | 0.21 | 0.2 | 3.53 | 17.56 |

Note: Min: Minimum; Max: Maximum; M: Median; SD: Standard Deviation; Mdn: Median; * hourly connection time; ** hourly connection time as a function of an SCT; $n = 251$.

We evaluated the differences in the level of Technological Acceptance of the teachers, considering the scientific area where they work (OECD area). We did not find statistically significant differences in teachers' scores for each group (F (5,49.42) = 1.78, $p = 0.13$). The same occurred when analyzing the dimensions of Perceived Usefulness (F (5,49.36) = 2.29, $p = 0.06$), and Perceived Ease (F (5,49.43) = 2.0489, $p = 0.09$).

Concerning academic analytics, we evaluated the teachers' time spent in the Canvas LMS, we found that teachers spent an average of 13.36 h in the virtual classroom for each credit (SCT) their subject had during the semester. However, since the standard deviation found in the data was substantial (SD = 17.82), we incorporated the median information indicating a dedication of Mdn = 8.0 h per credit (SCT) of the subject during the semester as a better indicator.

We evaluated the possible differences with respect to the weighted time on the platform according to the OECD area where the teachers work. We found statistically significant differences between groups (F (5,52.01) = 3.23, $p < 0.05$). The Lincoln posthoc test indicated significant differences between teachers working in Agricultural Sciences (M = 10.37; SD = 9.48) vs. Engineering and Technology (M = 30.62; SD = 35.67; $p < 0.05$), between Natural Sciences (M = 10.8; SD = 9.93) vs. Engineering and Technology (M = 30.62;

SD = 35.67; *p* < 0.05). Between Social Sciences (M = 11.15; SD = 14.56) vs. Engineering and Technology (M = 30.62; SD = 35.67; *p* < 0.05), and between Humanities (M = 9.85; SD = 10.31) vs. Engineering and Technology (M = 30.62; SD = 35.67; *p* < 0.05). In the case of teachers in Medical and Health Sciences (M = 13.74; SD = 18.19), they showed trend differences with the group of teachers of Engineering and Technology (M = 30.62; SD = 35.67; *p* = 0.051). Teachers in the Engineering and Technology spent more time on the LMS than teachers working in other scientific areas. With respect to the average time of the sessions, we found no statistically significant difference between the groups of teachers according to OECD area (F (5,49.11) = 2.83, *p* = 0.06).

When analyzing the teachers' scores in terms of high (average > 3) and low technology acceptance scores. we found that 38 teachers fell into the category of low technology acceptance of the Canvas LMS (see Table 4). In the case of the teachers with a low technology acceptance, most of them were male (*n* = 24). When confirming the existence of statistically significant differences. We found that teachers in this group had shorter connection times and shorter sessions than teachers with high technological acceptance. Finding answering H1. Another interesting finding was that the teachers in this group were older than the teachers with high acceptance of the use of the Canvas LMS.

**Table 4.** Categorization of teachers considering the level of technological acceptance with the Canvas LMS.

| Variables | Acceptation Technological | | | | Yuen Test | |
| | Low *n* = 38 | | High *n* = 213 | | | |
| | M | SD | M | SD | T | AKP Effect |
| --- | --- | --- | --- | --- | --- | --- |
| Age | 52.45 | 10.15 | 47.41 | 11.19 | T(36.57) = 2.764 ** | 0.45 |
| Time of connection in the LMS | 7.51 | 13.22 | 14.40 | 18.35 | T(42.22) = 4.404 *** | 0.64 |
| Average session time | 0.23 | 0.27 | 0.27 | 0.27 | T(34.45) = 2.43 * | 0.41 |

Note: M: Median; SD: Standard Deviation; AKP effect: effect size; *n* = 251. * *p* < 0.05. ** *p* < 0.01. *** *p* < 0.001.

In Table 4 we show the connection time of teachers and the sociodemographic characteristics of each group. We evaluated the possible differences in the age of teachers according to the scientific area where they work (OECD area). We found statistically significant differences (F (5,51.44) = 4.71, *p* < 0.01). The Lincon posthoc test [62], indicated significant differences between teachers of Agricultural Sciences and teachers of Medical and Health Sciences (*p* < 0.01), teachers of Humanities (*p* < 0.05) and teachers of Engineering and Technology (*p* < 0.05). In this case, teachers who teach in Agricultural Sciences courses are older than the rest of the groups identified.

Regarding the relationship between the level of technological acceptance of the teacher (T1) and the connection time used by teachers in the LMS (T2). We found a positive and statistically significant association in the dimensions and the total of the TAM model (evidence answering H2). A moderate and positive correlation is observed with the perceived ease dimension (rho = 0.30. *p* < 0.001). Which indicates that the greater the perceived ease of use of the LMS. The greater the connection time of the teacher in the LMS (see Table 5).

**Table 5.** Spearman correlation on the relationship between technology acceptance and teacher and student analytics with the Canvas LMS.

| Variables | 1 | 2 | 3 | 4 | 5 | 6 | 7 | 8 |
|---|---|---|---|---|---|---|---|---|
| 1. Technological Acceptance | 1 | | | | | | | |
| 2. Perceived Ease | 0.87 *** | 1 | | | | | | |
| 3. Perceived Usefulness | 0.90 *** | 0.58 *** | 1 | | | | | |
| 4. Teacher's LMS connection time | 0.24 *** | 0.30 *** | 0.15 * | 1 | | | | |
| 5. Teacher's average session time | 0.14 * | 0.13 * | 0.12 * | 0.54 *** | 1 | | | |
| 6. Academic Performance | 0.15 * | 0.10 | 0.18 ** | −0.06 | −0.07 | 1 | | |
| 7. Percentage of student resources viewed | 0.20 ** | 0.26 *** | 0.09 | 0.43 *** | 0.20 ** | −0.03 | 1 | |
| 8. Teacher's age | −0.22 *** | −0.33 *** | −0.08 | −0.27 *** | −0.08 | −0.02 | −0.22 *** | 1 |

Note: ** $p < 0.05$. ** $p < 0.01$. *** $p < 0.001$.

### 3.2. Technological Acceptance to Teacher's LMS Canvas and Its Relationship to the Percentage of Resources Viewed and Student Academic Achievement in the End ERT by COVID-19

To respond to the second objective (see Section 1.3). We calculated the average percentage of resources viewed by students. Which was 45.64% (SD = 20.23). Regarding the usefulness and ease perceived by teachers about the Canvas LMS. A moderate and positive correlation was observed between the ease perceived by the teacher and the percentage of resources viewed by students (rho = 0.26. $p < 0.001$). Regarding teacher-perceived usefulness, it had a slight correlation with students' academic performance (rho = 0.18. $p < 0.01$). These results respond to hypotheses H3 and H4 raised in this study (see Table 5).

## 4. Discussion

Remote education in emergencies prompted teachers to incorporate virtual tools in their teaching processes as the only alternative to provide continuity in higher education training. One of the most used virtual tools by higher education institutions were LMSs. Although they had been incorporated before the pandemic, their use was scarce among both teachers and students. The purpose of this study was to analyze the relationship between the level of technological acceptance of teachers at the beginning of the ERT semester (T1) with (1) the time spent by teachers in the Canvas LMS, (2) the percentage of resources viewed their students in the LMS, and (3) the final academic achievement of the same students at the end of that semester (T2).

### 4.1. Technological Acceptance of the Canvas LMS and its Relationship to Time Spent Teacher during ERT by COVID-19

In general terms, teachers express a good level of technological acceptance regarding using the LMS Canvas for their courses. University teachers perceive the LMS as a tool of low difficulty, wherein they have learned to use it in an intuitive and self-taught way to conduct their online classes during the ERT. In addition, they perceive that this tool has been useful to develop their teaching, facilitating the completion of pedagogical tasks in less time.

This information indicates that the choice of LMS to implement online education in an untimely manner and without prior preparation was a wise decision by higher education institutions. Furthermore, LMSs have a user-friendly design that allows teachers to adapt to them quickly. Although various research [65,66] point out that the ERT does not meet the standards of proper online education, it appears that the teachers were able to carry out their classes thanks to the LMS that provided them with the appropriate functionalities. This was enough for them to achieve the objective of giving continuity to their academic programs during a first adaptation period.

When teachers perceive the LMS as easy to learn and use, they spend more time working on it. This finding is congruent with the theory about TAM since teachers' perception of LMS use is influenced by their acceptance of the technology [16–18] and these results respond to hypotheses H1 raised in this study.

Therefore, it would be necessary for higher education institutions to generate strategies that favor the perception of ease of use of the LMS, such as tutorials, help desks, and short training sessions. In this way, by resolving quickly and efficiently the obstacles that teachers face when learning how to use an LMS, the institutions would be promoting its use for teaching and avoiding the use of alternative tools outside the LMS. Although using tools outside the LMS may have advantages, they have the significant disadvantage of not having a system for tracking and storing data. making it challenging to make pedagogical decisions based on learning analytics.

LMS offers various tools in the same virtual environment, such as discussion forums, assessments with various types of questions, structuring of the virtual classroom, enriched text editors, and external tools. These tools make it possible to design learning paths that seek to respond to the needs of students [67]. LMS also allow the implementation of active learning methodologies, which could be sustainable learning spaces, in which students could go beyond conceptual knowledge, encouraging critical and socially responsible thinking in favor of society and industry [50]. The diversity of tools available in LMS favors teachers' attempts to integrate the synchronous with the asynchronous, delivering a complete learning experience for the student, where both modalities complement each other. Therefore, a teacher who perceives the LMS as easy to learn will positively integrate virtual tools into his or her pedagogical practices, which will positively impact student learning. Thus, in the institutional choice of the LMS, it is essential to consider the simplicity of the interface that teachers will face for the development of the educational task since the ease of user interaction with the interface plays a fundamental role in teachers' perception of the platform [24,25].

Consistent with other research, we found that older teachers have a lower level of technological acceptability [68]. Therefore, the time on the platform is substantially less than that of younger teachers. This result reflects the generation gap regarding the use of educational technologies [69].

On the other hand, we evaluated differences in the variables studied according to the scientific area where the teachers teach. Concerning technological acceptance, we found no differences. But we did find differences regarding the time spent by teachers in the LMS. In this case, teachers in the engineering and technology area spent more time than other teachers. These differences show the need to provide differentiated and focused support to the teachers.

Despite this, we were unable to assess teachers' level of experience and familiarity with the LMS before the ERT. It is important to consider how the acceptance of LMS during the COVID-19 pandemic differs between teachers with and without previous experience, since teachers with experience with these platforms, presented greater behavioral intention to use them. Consequently, it has been reported that for teachers with little experience, institutional support is important to encourage their use [14]. Thus, it is crucial to take institutional actions that promote less resistance and greater enjoyment in using virtual tools, particularly regarding the use of LMSs, especially considering the projection of a post-pandemic blended learning education [70,71].

*4.2. TechnologicalAacceptance to Teacher's LMS Canvas and Its Relationship to the Percentage of Resources Viewed and Student Academic Achievement in the End ERT by COVID-19*

From the students' perspective, they visualize more virtual classroom resources when their teacher perceives the LMS as easy to learn and use. Furthermore, students achieve better academic performance (GPA) when their teachers perceive the greater usefulness of the LMS to develop teaching and promote better learning in them. These findings point to the importance of attitudes and beliefs that teachers transmit to their students about the LMS in developing their online courses [55]. For example, if a teacher complains or devalues the functionalities of the LMS by conveying to the student that they are not very useful and challenging to use, they are probably discouraging its use. In the same context, the fact that the teacher's perceived usefulness of the LMS is related to better performance, even if the relationship is small, it indicates that the LMS could be promoting more effective

teaching practices to improve student performance. In the future, routine measurements should be conducted to understand the LMS-related needs of teachers to offer specific training considering the characteristics of the courses and the teachers who teach them [43].

The strength of the results of this research lies in the contribution that they can offer to the investigation of technological acceptance in university teachers, a context in which few studies have been developed [22]. The results also give more insight into how teacher data can be analyzed to improve the learning and teaching processes. In our case, we analyzed teacher analytics provided by LMS, an analysis that has been developed to a limited extent [67]. Using information from academic and learning analytics is relevant for research on the incorporation of technology in teaching and institutional decision-making that can facilitate this process [36,37].

However, the present study had several limitations:

1.  The sampling used to select participants limited the homogeneous distribution of participants in the groups of high and low technological acceptance; therefore, it was not possible to establish robust conclusions about the characteristics of teachers according to each group and the relationship of these with student variables.
2.  In terms of analytics, only one student indicator was evaluated (percentage of resources viewed).
3.  Due to the heterogeneity of the courses, it was not possible to identify the type of resource provided by the teacher in the LMS.
4.  The relationships found between the study variables could be affected by other teacher variables, such as, for example, previous experience with online education tools or LMS, the pedagogy used to teach the courses, the number of students or characteristics of the courses.

Future research could evaluate how the teacher's level of technological acceptance is related to the use of the different tools offered by the LMS. Likewise, it should be studied how the diversity of resources used for teaching and the type of course designs in the virtual environment intervene in the educational experience and the academic performance of students. For this, it is important to consider more detailed analyses using learning analytics and academic analytics. In addition, it could be evaluated how the personal characteristics of teachers (gender, age) and other variables linked to online education (level of experience, knowledge, and training) could intervene in the level of technological acceptance of teachers in higher education.

## 5. Conclusions

From the results found, we conclude that most teachers indicated having high beliefs of technological acceptance of the LMS Canvas during the ERT due to COVID-19. They report higher beliefs of ease compared to usefulness of the tool; these findings were similar in teachers from various scientific areas.

A small percentage of teachers reported low technological acceptance; these teachers were less connected to the Canvas LMS and were older. We also concluded that teachers' connection times varied according to the scientific area in which they teach. Teachers in the Engineering and Technology spent more hours on the LMS. However, when we compared this time considering the SCT of the subjects taught, we found no differences, which allows us to conclude that the characteristics of the courses could be variables that moderate the time spent by teachers in the LMS.

We conclude that the greater the teacher's perceived ease of use, the higher the percentage of resources viewed by students, and in a weak but significant way, the greater the teachers' perception of usefulness, the higher the academic performance of their students.

University authorities have an important role in the implementation of tools and policies that promote the quality of education, as well as the accompaniment of teachers during these processes. These aspects allow the promotion of an effective and meaningful pedagogical practice by teachers during the development of online education. Teachers can be transforming agents of teaching methods and generate environments that allow

active participation of students and an environment that responds to their needs within a socially responsible and sustainable institutional framework.

**Supplementary Materials:** The following are available online at https://www.mdpi.com/article/10.3390/su132112167/s1, Table S1. Matrix of information extracted from the papers selected in the review.

**Author Contributions:** Conceptualization. R.C.-R. and K.L.P.; methodology. R.C.-R. and J.M.-N.; software. J.M.-N. and F.P.; validation. J.M.-N. and F.P.; formal analysis. R.C.-R., K.L.P. and N.C.S.M. investigation. R.C.-R., K.L.P. and N.C.S.M.; resources. K.L.P. and N.C.S.M.; data curation. J.M.-N. and F.P.; writing—original draft preparation. R.C.-R., K.L.P. and N.C.S.M.; writing—review and editing. R.C.-R., K.L.P. and N.C.S.M.; visualization. J.M.-N.; supervision. K.L.P. and N.C.S.M.; project administration. R.C.-R. and K.L.P.; funding acquisition. K.L.P. and N.C.S.M. All authors have read and agreed to the published version of the manuscript.

**Funding:** This research was funded by Unidad de Fortalecimiento Institucional of the Ministerio de Educación Chile. project InES 2018 UCO1808 Laboratorio de Innovación educativa basada en investigación para el fortalecimiento de los aprendizajes de ciencias básicas en la Universidad de Concepción.

**Institutional Review Board Statement:** The study was conducted according to the guidelines of the Declaration of Helsinki and approved by the Institutional Ethics Committee of University of Concepción (protocol code CEBBE-656-2020, date of approval April 2020).

**Informed Consent Statement:** Informed consent was obtained from all subjects involved in the study.

**Data Availability Statement:** Further inquiries can be directed to the corresponding author/s.

**Conflicts of Interest:** The authors declare no conflict of interest.

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
