# Peer review of "Longitudinal Analysis of Teacher Technology Acceptance and Its Relationship to Resource Viewing and Academic Performance of College Students during the COVID-19 Pandemic"

_sustainability, doi:10.3390/su132112167_

Round 1
Reviewer 1 Report
This article is well written and presents some useful ideas for further research.
Some unconventional and incomplete sentencing needs to be remedied in the abstract, although word count is probably an issue. For example, "Simple prospective design." Provide the full word or phrase - T1 and T2 need to be explained briefly.
The introduction was clear with relevant terms explained and literature introduced in a logical structure to contextualise the study and argue the need for the current research.
Line 124 - correct this: "they did not neglected"
Line 159 - correct this ""enable to improve"
Lines 167 and 168 - correct this "Moreover, there is research that report that teachers perceived ease of use and user satisfaction; The features reported as relevant of this LMS are the possibilities of structuring the class"
Line 176 - correct "of acceptance technology also allows to evaluate"
Line 180 correct "some articles shown"
Time spent and level of acceptance - expected
Discussion - Links between student achievement and teacher acceptance of LMS are associations only - they are not causal. For example, student achievement could be higher or lower because of factors not directly related to the LMS - teacher pedagogies, class size, use of formative assessment, and assessment type. This and other factors like this need to be acknowledged in limitations. Perhaps a discussion of implications for future research involving more nuanced or fine-grained analyses would also be helpful here.
Critical consideration of aspects such as prior familiarity with the LMS and prior experience/exposure, for example, at a different institute prior to the pandemic - this needed to be considered.
Author Response
Dear Reviewer,
We are writing relating to manuscript “Teacher Technology Acceptance and its relationship to resource viewing and academic performance of college students during the COVID-19 pandemic”. Authors: Rubia Cobo-Rendon, Karla Lobos Peña, Javier Mella-Norambuena, Nataly Cisternas San Martin & Fernando Peña.
We would like to thank for their comments, which, we believe, have greatly improved the quality of our article. We have modified the text accordingly and respond to each of the issues raised below.
Reviewer's comments
- This article is well written and presents some useful ideas for further research. Some unconventional and incomplete sentencing needs to be remedied in the abstract, although word count is probably an issue. For example, "Simple prospective design." Provide the full word or phrase - T1 and T2 need to be explained briefly.
Thank you very much for the suggestion. We have rewritten the summary in order to make it more understandable for the readers (see abstract)
- The introduction was clear with relevant terms explained and literature introduced in a logical structure to contextualise the study and argue the need for the current research.
This is a coment
- Line 124 - correct this: "they did not neglected"
Thank you very much, we have modified the wording of the sentence
Line 159 - correct this ""enable to improve"
Thank you very much, we have modified the wording of the sentence
- Lines 167 and 168 - correct this "Moreover, there is research that report that teachers perceived ease of use and user satisfaction; The features reported as relevant of this LMS are the possibilities of structuring the class"
Thank you very much, we have modified the wording of the sentence
- Line 176 - correct "of acceptance technology also allows to evaluate
Thank you very much, we have modified the wording of the sentence
- Line 180 correct "some articles shown"
Thank you very much, we have modified the wording of the sentence
- Time spent and level of acceptance – expected
Thank you very much, we have modified the wording of the sentence
- Discussion - Links between student achievement and teacher acceptance of LMS are associations only - they are not causal. For example, student achievement could be higher or lower because of factors not directly related to the LMS - teacher pedagogies, class size, use of formative assessment, and assessment type. This and other factors like this need to be acknowledged in limitations. Perhaps a discussion of implications for future research involving more nuanced or fine-grained analyses would also be helpful here.
Thank you very much for the suggestion. We have described, in the limitations section, the effect that other variables not considered in this research could have on the academic performance of the students.
We suggest conducting other research that considers specific variables (e.g., type of resource) in the section on future studies.
- Critical consideration of aspects such as prior familiarity with the LMS and prior experience/exposure, for example, at a different institute prior to the pandemic - this needed to be considered.
Thank you very much for the suggestion. We have described in the limitations section, the possible effect of variables such as familiarity or previous knowledge with the LMS on the findings found.

Reviewer 2 Report
The objectives of the study are interested and current. However, it seems too local and restricted regarding the variables analyzed. This could present some bias that make not publicable in the way it is. I give authors some feedback:
Introduction
Line 59: “The TAM model has been undisputedly the most tested and validated model in different contexts and studies, confirming its robustness and ability to predict technology 60 adoption in users [14].” However the reference presented to support this affirmation came from proceedings in a conference. It would be necessary to contribute to this sentence with studies published in high value journals, or at least with double blind review processes.
Line 133-134: I don’t understand the reason of this sentence here, there is no contribution to the reading of the article. In point 1.3 authors explain the objectives of the study.
Materials and Methods
Some more information is required about the sample: which were the universities that participated in the study? Connecting with this question is this study local and restricted to a specific university?. If there were more universities involved, which were their characteristics? (private or public ones, degrees and programs, …). Furthermore, which were the characteristics of teachers (full or partial time), background. There are different factors here that could also influence the technological acceptance, specially the background of the teachers.
Authors include the consistency of the instrument, English version. If a Spanish version was used, has been validated? It seems that adequate consistency is found for the study, but a reference about the Spanish validation is needed.
Are credits ECTS credits? Could the authors explain how the credits are calculated?
Results
I suggest authors to analyse the results according the background and experience of teachers in technology systems. If this is not taken into account there is a serious risk to commit bias so results would not be valid.
Discussion
The study is interested by in my opinion there is a lack of variables to take into account. There is emphasis on the differences according the age of the teachers but as I remark the background and experience of teachers is a clue regarding the technological acceptance. And this is not considered or incorporated in some way in the study. Because of the possible bias I mentioned it is difficult to understand the usefulness of the study.
Authors said in the discussion “Furthermore, they perceive that this tool has 354 been useful to develop their teaching since it increases the productivity and effectiveness 355 of their work and optimizes the investment of time.” It is not clear according the results presented how the perception of productivity and effectiveness is measured. Technological acceptance does not mean directly best productivity or effectiveness. This assumption should be proven or support with other studies.
It is really interested the discussion about the relationship between time working and the perception of LMS as easy to learn. In fact, this can be an input to design actions and training programs for teachers.
References
Important information is missed or there are errors in some references. I only indicate some of them: numbers 4, 14, 20, 42, 48. I advise you to review all of them according the guidelines.
Author Response
Dear Reviewer,
We are writing relating to manuscript “Teacher Technology Acceptance and its relationship to resource viewing and academic performance of college students during the COVID-19 pandemic”. Authors: Rubia Cobo-Rendon, Karla Lobos Peña, Javier Mella-Norambuena, Nataly Cisternas San Martin & Fernando Peña.
We would like to thank for their comments, which, we believe, have greatly improved the quality of our article. We have modified the text accordingly and respond to each of the issues raised below.
Reviewer's comments
- The objectives of the study are interested and current. However, it seems too local and restricted regarding the variables analyzed. This could present some bias that make not publicable in the way it is. I give authors some feedback:
Introduction
Line 59: “The TAM model has been undisputedly the most tested and validated model in different contexts and studies, confirming its robustness and ability to predict technology 60 adoption in users [14].” However, the reference presented to support this affirmation came from proceedings in a conference. It would be necessary to contribute to this sentence with studies published in high value journals, or at least with double blind review processes.
Thank you very much for the suggestion. We have incorporated other references that support the idea described above (see lines 60-62)
- Line 133-134: I don’t understand the reason of this sentence here, there is no contribution to the reading of the article. In point 1.3 authors explain the objectives of the study.
Thank you very much for the suggestion. We have deleted the sentence in the manuscript
- Materials and Methods
Some more information is required about the sample: which were the universities that participated in the study? Connecting with this question is this study local and restricted to a specific university?. If there were more universities involved, which were their characteristics? (private or public ones, degrees and programs, …). Furthermore, which were the characteristics of teachers (full or partial time), background. There are different factors here that could also influence the technological acceptance, specially the background of the teachers.
Thank you very much for the suggestion. We have indicated in the participant's section that the study corresponds to data from a university in Chile. We have also made a more detailed description of the participating teachers. Furthermore, we include a table with the scientific area where they work, the number of years of teaching experience within the university and the type of dedication (See lines 241-251)
- Authors include the consistency of the instrument, English version. If a Spanish version was used, has been validated? It seems that adequate consistency is found for the study, but a reference about the Spanish validation is needed.
Thank you very much for the suggestion. We incorporated the psychometric information previously analyzed for the manuscript (see lines 281-284)
- Are credits ECTS credits? Could the authors explain how the credits are calculated?
SCT are the equivalent of ECTS in Chile. SCT credits are designed based on the estimated time dedicated by the student to achieve the learning outcomes. One SCT credit is equivalent to a range of workloads between 24 and 31 hours by the student. This information was it incorporated in the description of variable's total time on the platform their teacher in the method section (see lines 297-300)
- Results
I suggest authors to analyse the results according the background and experience of teachers in technology systems. If this is not taken into account there is a serious risk to commit bias so results would not be valid.
Thank you very much for the suggestion. We have included the analysis of the variables considering the scientific areas where the teachers work (background) (See lines 372-376 and 383-397). We did not consider previous experience as a research variable, so we do not have that information. However, we include an analysis of this in the discussion at the end of section 4.1. and we include it as a limitation of this work
- Discussion
The study is interested by in my opinion there is a lack of variables to take into account. There is emphasis on the differences according the age of the teachers but as I remark the background and experience of teachers is a clue regarding the technological acceptance. And this is not considered or incorporated in some way in the study. Because of the possible bias I mentioned it is difficult to understand the usefulness of the study.
Thank you very much for the suggestion. We have discussed the differences by area where teachers work in terms of technological acceptance and described the importance of considering the characteristics of teachers in the approach of universities to the use of tools for online education (see 502-512).
- Authors said in the discussion “Furthermore, they perceive that this tool has 354 been useful to develop their teaching since it increases the productivity and effectiveness 355 of their work and optimizes the investment of time.” It is not clear according the results presented how the perception of productivity and effectiveness is measured. Technological acceptance does not mean directly best productivity or effectiveness. This assumption should be proven or support with other studies.
Thank you very much for the suggestion. We have modified the wording of this information, considering the content of the TAM items (see lines 462-464)
- It is really interested the discussion about the relationship between time working and the perception of LMS as easy to learn. In fact, this can be an input to design actions and training programs for teachers.
Thank you very much for your comment. We appreciate your feedback
- References
Important information is missed or there are errors in some references. I only indicate some of them: numbers 4, 14, 20, 42, 48. I advise you to review all of them according the guidelines.
Thank you very much for the suggestion. We have reviewed in detail the references of the manuscript.

Reviewer 3 Report
In line 63, instead of "his", I suggest you use "his / her".
The authors of the article never used the word "sustainability", although the article was submitted to the scientific journal Sustainability. In this connection, I would like to ask what is the relationship between the topic and research results presented and sustainability. What exactly was assessed in terms of sustainability? The word "sustainable" was used only once in the article, but it was in the Introduction, so it is not a direct contribution of the Authors to the development of knowledge on sustainability. Once the word "sustainable" was included in the link in References (line: 527), but this also does not mean that the presented topic solves the problems of sustainability. Personally, I think that the article can tackle the issue of sustainability, and it can be expressed at the stage of formulating the research problem (summarizing the considerations made in the Introduction), or even for the purpose of research.
In general, I suggest that the Authors formulate in a separate paragraph the research problem they undertake to solve.
The article uses the terms "technology" and "technological" many times to refer to the teacher and educational activities. In my opinion, in the beginning of the article, it is necessary to clearly define the concept of technology in terms of the considerations undertaken regarding the tasks carried out in education / higher education. Of course, it can be assumed that everyone knows (or should know) what technology is and what it covers. However, the term "technology" refers to many areas of life, it is related to agriculture and industry, there is communication and information technology. And this is not the same technology as the educational area, so you need to define it in detail.
I am not sure if "deepening the knowledge" about the acceptance of technology by the teacher (lines: 133-134) is sufficiently scientific goal of the undertaken research. The aim of the work could be formulated in a more scientific way. Therefore, as I mentioned before, it is important to correctly formulate the research problem from which the scientific purpose of the research / work results. In general, in lines 290-292, the Authors mentioned the purpose of the work once again, which differs significantly from the wording of the goal of work in lines: 133-134. On line 293 the authors wrote: "… we will present the results according to the specific objectives previously described." I cannot find where the Authors listed specific objectives previously. Please indicate this place or complete the information. If specific objectives are listed at the end of section 1.3, it must be explicitly stated that they are specific objectives. Meanwhile, the authors wrote: "Therefore, in this study we specifically sought to:". Such ambiguous phrases make the article very difficult to read and create unnecessary ambiguities.
When formulating the research problem, it would be worth mentioning that the development of education, especially in the times of a pandemic, is associated with the search for alternative solutions for assessing the process of educating students. Here you can, for example, refer to the article "The topic of the ideal dairy farm can inspire how to assess knowledge about dairy production processes: A case study with students and their contributions".
If the Authors use the term "level of technological acceptance" (eg in line: 185), then in what range does "level" fall, ie what is its lowest and highest value? Please reply.
If the Authors use the term "level of teacher technology acceptance" (eg in line: 188-189), what range does "level" fall into, i.e. what is its lowest and highest value? Please reply.
No dots are put at the end of chapter / subsection titles (e.g. line: 172).
In my opinion, in Section 2.1 (Participants), the participants in the study, especially the teachers, need to be described in more detail. If the average age of teachers and the standard deviation (SD) are given, it would also be useful to mention the age range (the age of the youngest and oldest teachers participating in the study). In the case of students, the standard deviation (SD) is much smaller, so the limit values for the age of students do not need - in my opinion - to be specified. The more important issue in the description of the research group - teachers, is their experience. Therefore, I think that it is necessary to fill in the descriptive statistics what has been the working time (in years) (mean ± SD) of the teachers participating in the study so far. I believe that this factor may be crucial for the evaluation of the research results and their discussion. Besides, I did not find out what subjects the teachers were teaching from the description of the research group. Were they general or specialist subjects? Were they technical, humanities or natural sciences (life sciences), or maybe medical, economic or musical? The same remarks apply to the group of students participating in the study. What faculties did the students study in? The description contained only laconic information that over 22% were first-year students. This, in my opinion, is not enough. I think that the above-mentioned missing information about teachers and students may have a significant impact on the obtained research results and be useful in a fuller discussion of the research results.
In my opinion, point 2.4 was incorrectly titled. This section describes a statistical approach to compiling survey results, so I suggest a title: Statistical analysis.
The data summarized in Table 1 is very badly analyzed. This is because the units of each quantity are not given. For example, if "Platform connection time" is specified then units must be added in an additional column, in this case h / credit, as written on lines 303-304. The same applies to the other terms in Table 1. Without specifying the units, it is not known what the individual mean values mean. In addition, in Table 1, numbers are given with commas, while in English, dots should be used when writing the number with decimal places.
The abbreviations M, SD, T, AKP effect need to be explained in Table 2. Of course, most readers should know what these acronyms are about, but probably not all. In addition, the first column must specify the units in which individual variables are expressed.
I think a chapter Conclusions could be useful at the end of the article. Thanks to this, it would be possible to efficiently summarize the presented research.
Some publications in References should be corrected, i.e. journal abbreviations should be included.
Author Response
Dear Reviewer,
We are writing relating to manuscript “Teacher Technology Acceptance and its relationship to resource viewing and academic performance of college students during the COVID-19 pandemic”.
We would like to thank for their comments, which, we believe, have greatly improved the quality of our article. We have modified the text accordingly and respond to each of the issues raised below.
Reviewer's comments
- In line 63, instead of "his", I suggest you use "his / her".
Thank you very much, we have modified the wording of the sentence
- The authors of the article never used the word "sustainability", although the article was submitted to the scientific journal Sustainability. In this connection, I would like to ask what is the relationship between the topic and research results presented and sustainability. What exactly was assessed in terms of sustainability? The word "sustainable" was used only once in the article, but it was in the Introduction, so it is not a direct contribution of the Authors to the development of knowledge on sustainability. Once the word "sustainable" was included in the link in References (line: 527), but this also does not mean that the presented topic solves the problems of sustainability. Personally, I think that the article can tackle the issue of sustainability, and it can be expressed at the stage of formulating the research problem (summarizing the considerations made in the Introduction), or even for the purpose of research.
Thank you very much for the suggestion. We have rewritten the problem formulation section, with more emphasis on the research contribution to sustainability (see lines 179-232).
We have also incorporated some elements in the discussion of the paper.
- In general, I suggest that the Authors formulate in a separate paragraph the research problem they undertake to solve.
Thank you very much for the suggestion. We have rewritten the problem formulation section.
- The article uses the terms "technology" and "technological" many times to refer to the teacher and educational activities. In my opinion, in the beginning of the article, it is necessary to clearly define the concept of technology in terms of the considerations undertaken regarding the tasks carried out in education / higher education. Of course, it can be assumed that everyone knows (or should know) what technology is and what it covers. However, the term "technology" refers to many areas of life, it is related to agriculture and industry, there is communication and information technology. And this is not the same technology as the educational area, so you need to define it in detail.
Thank you very much for the suggestion. We have described the information requested in the introduction in section 1.1. (see lines 78-87)
- I am not sure if "deepening the knowledge" about the acceptance of technology by the teacher (lines: 133-134) is sufficiently scientific goal of the undertaken research. The aim of the work could be formulated in a more scientific way. Therefore, as I mentioned before, it is important to correctly formulate the research problem from which the scientific purpose of the research / work results. In general, in lines 290-292, the Authors mentioned the purpose of the work once again, which differs significantly from the wording of the goal of work in lines: 133-134. On line 293 the authors wrote: "… we will present the results according to the specific objectives previously described." I cannot find where the Authors listed specific objectives previously. Please indicate this place or complete the information. If specific objectives are listed at the end of section 1.3, it must be explicitly stated that they are specific objectives. Meanwhile, the authors wrote: "Therefore, in this study we specifically sought to:". Such ambiguous phrases make the article very difficult to read and create unnecessary ambiguities.
Thank you very much for the suggestion. We have rewritten the problem formulation section and have specifically stated the research objectives and hypotheses.
- When formulating the research problem, it would be worth mentioning that the development of education, especially in the times of a pandemic, is associated with the search for alternative solutions for assessing the process of educating students. Here you can, for example, refer to the article "The topic of the ideal dairy farm can inspire how to assess knowledge about dairy production processes: A case study with students and their contributions".
Thank you very much for the suggestion we have incorporated the ideas raised in the problem formulation section.
- If the Authors use the term "level of technological acceptance" (eg in line: 185), then in what range does "level" fall, ie what is its lowest and highest value? Please reply. If the Authors use the term "level of teacher technology acceptance" (eg in line: 188-189), what range does "level" fall into, i.e. what is its lowest and highest value? Please reply.
Thank you very much for the suggestion. We have incorporated in the description of the instrument the interpretation of the levels of technological acceptance and in table 3, which describes the scores, we have added the minimum and maximum scores achieved by the participants (see table 3)
- No dots are put at the end of chapter / subsection titles (e.g. line: 172).
We have deleted the final dot
- In my opinion, in Section 2.1 (Participants), the participants in the study, especially the teachers, need to be described in more detail. If the average age of teachers and the standard deviation (SD) are given, it would also be useful to mention the age range (the age of the youngest and oldest teachers participating in the study). In the case of students, the standard deviation (SD) is much smaller, so the limit values for the age of students do not need - in my opinion - to be specified. The more important issue in the description of the research group - teachers, is their experience. Therefore, I think that it is necessary to fill in the descriptive statistics what has been the working time (in years) (mean ± SD) of the teachers participating in the study so far. I believe that this factor may be crucial for the evaluation of the research results and their discussion. Besides, I did not find out what subjects the teachers were teaching from the description of the research group. Were they general or specialist subjects? Were they technical, humanities or natural sciences (life sciences), or maybe medical, economic or musical? The same remarks apply to the group of students participating in the study. What faculties did the students study in? The description contained only laconic information that over 22% were first-year students. This, in my opinion, is not enough. I think that the above-mentioned missing information about teachers and students may have a significant impact on the obtained research results and be useful in a fuller discussion of the research results.
Thank you very much for your suggestions. We have expanded the description of the teachers and students in the participant’s section. In the case of teachers, we also evaluated the existence of statistically significant differences with respect to age, level of technological acceptance, time on the platform and time in sessions with the LMS. We describe the findings in the results and comment on the new information generated in the discussion. (See 240-264).
- In my opinion, point 2.4 was incorrectly titled. This section describes a statistical approach to compiling survey results, so I suggest a title: Statistical analysis.
Thank you very much for the suggestion. We changed the name of the section
- The data summarized in Table 1 is very badly analyzed. This is because the units of each quantity are not given. For example, if "Platform connection time" is specified then units must be added in an additional column, in this case h / credit, as written on lines 303-304. The same applies to the other terms in Table 1. Without specifying the units, it is not known what the individual mean values mean. In addition, in Table 1, numbers are given with commas, while in English, dots should be used when writing the number with decimal places.
Thank you very much for the suggestion. We have made the requested modification (see table 3)
- The abbreviations M, SD, T, AKP effect need to be explained in Table 2. Of course, most readers should know what these acronyms are about, but probably not all. In addition, the first column must specify the units in which individual variables are expressed.
Thank you very much for the suggestion. We have made the requested modification (see table 4)
- I think a chapter Conclusions could be useful at the end of the article. Thanks to this, it would be possible to efficiently summarize the presented research.
Thank you very much for the suggestion. We have made the requested modification (see lines 571-593)
- Some publications in References should be corrected, i.e. journal abbreviations should be included.
Thank you very much for the suggestion. We have reviewed in detail the references of the manuscript.

Round 2
Reviewer 2 Report
Dear authors,
I think the article has improved significantly. Thank you very much for taking in consideration my suggestions. Now, objectives and hypothesis are clearer, also the context of the study. Results and discussion are better structured and deeper. Readers will be able to extract information of interest if they wish to apply this methodology to other contexts and use the information obtained in the design of academic strategies.
Even so, I still have a suggestion for improvement. In my previous review I asked for more explanation on line 59 (references an so). Now, authors have completed this in lines 60-62. I have still some doubts regarding this paragraph. The authors say: “This theory is based on the premise that the reaction and perception of a person will determine his/her attitude and behavior, i.e., reactions and perceptions regarding the use of a particular technology is influenced by the attitude towards the acceptance of these technologies [17].” The sentence is not clear, authors repeat the idea but which is the cause and which the consequence?, that is, do the reaction and perception determine the attitude or are they influenced by the attitude? I recommend the authors to rewrite this sentence. It is still confusing.
Author Response
Thank you very much for all your contributions. We have modified the wording of the paragraph you comment (see lines 63-66). It is now in the manuscript as follows: This model is explained from the Theory of Reasoned Action [16]. This theory is based on the premise that the perception of a person will determine his/her attitude and behavior, i.e., the attitude and behavior that a person may have about the use of a particular technology is influenced by the perceptions that the person has about it [17].Reviewer 3 Report
The abbreviation TAM used in Abstract requires a full name when first used. Therefore, the concept of Technology Acceptance Model (TAM) must be included in the Abstract. If the reader wants to learn the most important information about the research on the basis of Abstract, the lack of the full TAM name will be a significant limitation.
Author Response
Thank you very much for your input, we have modified the summary to include the full name of the TAM model.